# HDAC6-Selective Inhibitor Overcomes Bortezomib Resistance in Multiple Myeloma

**DOI:** 10.3390/ijms22031341

**Published:** 2021-01-29

**Authors:** Sang Wu Lee, Soo-Keun Yeon, Go Woon Kim, Dong Hoon Lee, Yu Hyun Jeon, Jung Yoo, So Yeon Kim, So Hee Kwon

**Affiliations:** College of Pharmacy, Yonsei Institute of Pharmaceutical Sciences, Yonsei University, Incheon 21983, Korea; tkddn407@naver.com (S.W.L.); ilove_oov@hanmail.net (S.-K.Y.); goun6997@daum.net (G.W.K.); tci30@naver.com (D.H.L.); hyun953@naver.com (Y.H.J.); jungy619@yonsei.ac.kr (J.Y.); ksy_dct@naver.com (S.Y.K.)

**Keywords:** HDAC6, bortezomib-resistance, HDAC6-selective inhibitor, bortezomib, carfilzomib, multiple myeloma, LMP2, combination therapy

## Abstract

Although multiple myeloma (MM) patients benefit from standard bortezomib (BTZ) chemotherapy, they develop drug resistance, resulting in relapse. We investigated whether histone deacetylase 6 (HDAC6) inhibitor A452 overcomes bortezomib resistance in MM. We show that HDAC6-selective inhibitor A452 significantly decreases the activation of BTZ-resistant markers, such as extracellular signal-regulated kinases (ERK) and nuclear factor kappa B (NF-κB), in acquired BTZ-resistant MM cells. Combination treatment of A452 and BTZ or carfilzomib (CFZ) synergistically reduces BTZ-resistant markers. Additionally, A452 synergizes with BTZ or CFZ to inhibit the activation of NF-κB and signal transducer and activator of transcription 3 (STAT3), resulting in decreased expressions of low-molecular-mass polypeptide 2 (LMP2) and LMP7. Furthermore, combining A452 with BTZ or CFZ leads to synergistic cancer cell growth inhibition, viability decreases, and apoptosis induction in the BTZ-resistant MM cells. Overall, the synergistic effect of A452 with CFZ is more potent than that of A452 with BTZ in BTZ-resistant U266 cells. Thus, our findings reveal the HDAC6-selective inhibitor as a promising therapy for BTZ-chemoresistant MM.

## 1. Introduction

Multiple myeloma (MM) is characterized by abnormally proliferating plasma cells derived from B cells [1] and ranks second as the cause of death from hematological malignancy [2]. Typical symptoms of MM include anemia, bone destruction, hypercalcemia, infection, and renal failure due to impaired immune function [3]. Recently, the survival rate of MM patients has improved due to the development of autologous stem cell transplantation and novel therapeutic agents that include proteasome inhibitors (PIs) and immunomodulatory drugs [4]. Despite recent advances in MM treatment, most patients fall into cyclic relapse and eventually develop refractory disease due to residual MM [5]. Many researchers have reported the recurrence and drug resistance in MM patients who have previously received chemotherapy, which remains a major obstacle for curing MM. Thus, because there are currently no effective therapies to treat chemotherapy-resistant MM, novel effective treatments for MM need to be identified.

Bortezomib (BTZ, Velcade) is the first therapeutic PI that was approved by the US Food and Drug (FDA) Administration in 2003 for the treatment of MM [6]. BTZ binds to the catalytic subunit of the proteasome with high affinity and inhibits the ubiquitin-proteasome system (UPS) [7,8]. Consequently, BTZ indirectly regulates the apoptotic pathway and cell cycle progression, affecting the downstream signaling of UPS [9]. The nuclear factor kappa B (NF-κB) pathway is a typical cell signaling regulated by the UPS. The NF-κB inhibitor (I-κB) masks the nuclear localization signal of NF-κB in the cytoplasm. When I-κB is degraded by the UPS, NF-κB translocates into the nucleus and transcribes its target genes, such as anti-apoptotic genes. However, BTZ inhibits activation of the NF-κB pathway by preventing the degradation of I-κB [10,11]. BTZ also activates both intrinsic and extrinsic apoptotic pathways [12]. In addition, BTZ causes an accumulation of the ubiquitin-conjugated proteins that enhances endoplasmic reticulum stress with cytotoxicity, resulting in apoptosis induction [13].

BTZ has been approved to treat newly diagnosed and relapsed and/or refractory (R/R) MM [14,15]. Although significant improvements in the management of MM with BTZ have been reported, relapses are common, and treatment efficacy is reduced in MM patients. One of the various reasons for BTZ resistance is the abnormal expression of proteasome subunits that BTZ targets [16]. BTZ preferentially binds to the β5 subunit of the proteasome. The catalytic subunits of constitutive proteasome, β5 subunit (known as *PSMB5*), are significantly upregulated, whereas β1 (encoded by *PSMB1*) and β2 (encoded by *PSMB2*) are modestly upregulated in cancers. However, no significant changes were seen in the mRNA levels of β5, so the posttranscriptional mechanism seems to upregulate the protein level of β5 [16]. Recently, *PSMB5* mutations were also found by deep sequencing in BTZ-treated MM patients [17]. In the immunoproteasome, low-molecular-mass polypeptide 7 (LMP7, iβ5, *PSMB8*), LMP2 (iβ1, *PSMB9*), and multicatalytic endopeptidase complex-like-1 (MECL1, iβ2, *PSMB10*) subunits are catalytically active subunits that BTZ targets [18,19]. Myeloma co-expresses the constitutive proteasome and immunoproteasome [20]. A recent study has shown that the expression of LMP7 and LMP2 is modulated via suppression of EGFR/JAK1/STAT3 signaling by tight junction protein 1 in BTZ-resistant MM [21]. Moreover, SCF^Skp2^ is upregulated in BTZ-resistant MM and promotes the degradation of p27^Kip1^, which inhibits cyclin-dependent kinase, indicating that SCF^Skp2^ is another biomarker of BTZ resistance [22]. Therefore, the discovery of biomarkers of BTZ resistance is crucial to overcome BTZ resistance.

Combined treatment of BTZ with other anti-myeloma agents is one of the therapeutic strategies for overcoming BTZ resistance. Histone deacetylase (HDAC) inhibitor is one promising approach that shows anti-myeloma effects in preclinical and clinical studies [23] and overcomes drug resistance. Suberoylanilide hydroxamic acid (SAHA, vorinostat) is the first class of pan-HDAC inhibitor approved by the US FDA in 2009 for cutaneous T cell lymphoma [24]. Another pan-HDAC inhibitor panobinostat (LBH589) was approved by the US FDA in 2015 for R/R MM in combination with BTZ and dexamethasone (DEX) [25]. The clinical trial data showed that the combination treatment of BTZ with vorinostat or panobinostat significantly showed responses in R/R MM patients [25,26,27,28]. However, pan-HDAC inhibitors are associated with significantly increased toxicity in this combination, which restricts their clinical utility. To minimize toxicity but maintain efficacy, the HDAC6-selective inhibitor, ACY-1215 (ricolinostat), was tested and showed anticancer effects in preclinical and clinical studies. Ricolinostat demonstrated anticancer activity in hematological malignancies, including MM, diffuse large B cell lymphoma, both germinal center B cell and activated B cell, follicular lymphoma, mantle cell lymphoma, and T cell lymphoma [27,29,30]. Synergistic effects have been demonstrated with ricolinostat and BTZ or carfilzomib (CFZ) in BTZ-sensitive MM [27,29,31]. Recently, Vogl et al. showed that the combination of ricolinostat with BTZ and DEX responds in patients with R/R MM (two-thirds were BTZ refractory) [32], and an ongoing trial (phase 1 and 2) is being explored (NCT 01323751).

In this study, we investigated the antimyeloma effects of HDAC6-selective inhibitors, A452 [33] and ACY-1215, in both BTZ-sensitive and BTZ-resistant MM cells. These results show that A452 and ACY-1215 efficiently inhibit the cell growth and viability in both BTZ-sensitive and BTZ-resistant MM cells. Overall, A452 is more cytotoxic than ACY-1215 in BTZ-resistant MM cells. We also observed that NF-kB, mitogen-activated protein kinase (MAPK), and protein kinase B (PKB, known as AKT) pathways involved in BTZ resistance were inactivated by HDAC6-selective inhibitor in both BTZ-sensitive and BTZ-resistant MM cells. Furthermore, treatment with HDAC6-selective inhibitor in BTZ-resistant MM cells not only reduced the activation of STAT3 and NF-kB but also decreased the expression of other BTZ-resistant markers, LMP2 and LMP7. Combination treatment of A452 and BTZ or CFZ synergistically reduced BTZ-resistant markers. Moreover, combination treatments with A452 and BTZ or CFZ restored the sensitivity of BTZ and synergistically induced apoptosis in BTZ-resistant MM cells. In summary, the HDAC6-selective inhibitor could overcome the BTZ resistance via inhibition of cell survival signaling and the STAT3–NF-κB–LMP2 pathway in BTZ-resistant MM.

## 2. Results

### 2.1. Establishment of BTZ-Resistant U266/VelR MM Cells

BTZ-resistant U266 MM cells (U266/VelR) have hyperphosphorylated NF-κB (p65) and ERK MAPK and are less sensitive to BTZ-induced cell cytotoxicity than BTZ-sensitive parental U266 cells [34]. To confirm and strengthen BTZ resistance, we performed a soft agar assay and isolated three BTZ-resistant U266 cell clones (U266/VelR-1, U266/VelR-2, and U266/VelR-3). Both BTZ-resistant U266/VelR-1 and U266/VelR-2 cells had hyper-phosphorylated NF-κB and ERK compared to the parental U266 cells (Figure 1A). Next, to confirm the BTZ resistance in both BTZ-resistant U266 cells, we examined the cell viability of both BTZ-resistant U266 cells after treatment with BTZ for 72 h. BTZ-resistant U266 MM cell lines showed a 10-fold higher resistance to BTZ compared with their sensitive counterparts (IC_50_ values 0.5 nM vs. 5 nM). While the viability of BTZ-sensitive U266 cells significantly decreased in a dose-dependent manner, both BTZ-resistant U266 cells showed BTZ resistance from 2 nM BTZ to 5 nM BTZ (Figure 1B). Additionally, we tested the activation of NF-κB and ERK by BTZ. The phosphorylated levels of p65 and ERK decreased dose dependently in BTZ-sensitive U266 cells, but not in either of the BTZ-resistant U266 cells (Figure 1C). At a higher concentration (10 nM BTZ), total levels of p65 and ERK were significantly reduced in both BTZ-sensitive and BTZ-resistant MM cells. In addition, the viability of both BTZ-sensitive and BTZ-resistant U266 cells similarly decreased at 10 nM BTZ (Figure 1B), undistinguishable at a higher concentration (10 nM BTZ). We observed an inverse relationship between the levels of BTZ-resistance markers and MM sensitivity to BTZ (Figure 1A,B). Therefore, these findings indicate that active NF-κB and MAPK pathways have key roles in BTZ resistance-related cell viability in MM cells.

### 2.2. A452 is More Cytotoxic than ACY-1215 in Both BTZ-Sensitive and BTZ-Resistant U266 Cells

A452 is a small-molecule inhibitor with a γ-lactam that selectively inhibits HDAC6 catalytic activity in various human cancer cells [33]. Recently, A452 demonstrated significant anticancer activity in solid tumors and blood cancers, including MM [35,36,37,38]. We, therefore, sought to determine whether A452 has antimyeloma activity in both BTZ-sensitive U266 and BTZ-resistant U266/VelR cells. First, we examined the cell growth and viability in both BTZ-sensitive and BTZ-resistant U266 cells after treatment with A452 for 72 h by cell counting kit-8 (CCK-8) assays (Figure 2). Regardless of the resistance of BTZ, A452 resulted in a time- and dose-dependent decrease of the cell growth and viability in both BTZ-sensitive and BTZ-resistant U266 cells with GI_50_ and IC_50_ values ~0.25 μM (Figure 2G). The viability of both BTZ-sensitive and BTZ-resistant U266 cells decreased to less than 40% when treated with 1 μM A452 (Figure 2B,D,F), but the viability decreased to 60% when treated with 1 μM ACY-1215, the only first-in-class clinically relevant small-molecule HDAC6 inhibitor (Appendix A). The IC_50_ value of ACY-1215 (IC_50_ values > 2.5 μM) was 10 times higher than that of A452 (~0.25 μM; Figure 2G), indicating that A452 is a more sensitive drug than ACY-1215 in both BTZ-sensitive and BTZ-resistant U266 cells. The value of GI_50_ was similar to IC_50_ in both BTZ-sensitive and BTZ-resistant U266 cells. Taken together, these findings indicate that BTZ-resistant MM cells are still sensitive to HDAC6 inhibitors and that both BTZ-sensitive and BTZ-resistant U266 cells are more sensitive to A452 than ACY-1215.

### 2.3. A452 in Combination with BTZ or CFZ Shows Synergistic Cytotoxicity in Both BTZ-Sensitive and BTZ-Resistant U266 Cells

The HDAC6-selective inhibitor A452 showed cytotoxicity in both BTZ-sensitive and BTZ-resistant U266 cells. Next, we investigated the combined effect of A452 and BTZ in BTZ-resistant U266/VelR and BTZ-sensitive U266 cell cytotoxicity. Cells were treated with A452, BTZ, and A452 in combination with BTZ in both BTZ-resistant and BTZ-sensitive U266 cells. The CCK-8 assay was performed to measure cell viability for 72 h. BTZ treatment alone had no distinct cytotoxic effects in BTZ-resistant U266 cells, whereas notable synergistic increases in cytotoxicity with A452 were observed (Figure 3; left panel). Next, we analyzed the synergism between A452 and BTZ by applying the Chou and Talalay method [39]. The combination of A452 and BTZ showed synergistic anti-MM activity with a combination index (CI) < 1.0 (Figure 3; right panel).

Next, we examined the combined cytotoxic effect of CFZ with A452 on BTZ-resistant U266/VelR and BTZ-sensitive U266 cells. The β5/β2 co-inhibition is known as the most effective cytotoxic in PI-sensitive and PI-resistant MM [20]. Among the available PIs, only high-dose CFZ provides β5/β2 co-inhibition. Although BTZ-resistant cells were partly cross-resistant to CFZ, which is structurally and mechanistically distinct from BTZ [40], these resistant cells remained sensitive to inhibition of proliferation by CFZ (Appendix A) [40,41] and reduced BTZ-resistant markers (Appendix A). In addition to the combination of A452 and BTZ, the cotreatment of A452 and CFZ showed synergistic anti-MM activity in both BTZ-sensitive and BTZ-resistant U266 cells (Appendix A). Therefore, these data suggest that A452 in combination with BTZ or CFZ synergistically induces cytotoxicity in both BTZ-sensitive and BTZ-resistant U266 cells.

### 2.4. A452 in Combination with BTZ or CFZ Synergistically Leads to Apoptosis in BTZ-Sensitive and BTZ-Resistant U266 Cells

PI-based MM therapy induces apoptosis due to excessive proteotoxic stress [42]. The combination treatment with A452 and other anticancer agents enhanced the efficacy of anticancer activity in solid tumors [37] and hematological malignancies [30,38]. However, the effect of combination treatment with A452 and BTZ or CFZ has not been studied in hematological malignancies. To characterize the mechanism of action of synergistic cytotoxicity induced by the combination treatment, we examined the activation of the apoptotic pathway in both BTZ-sensitive and BTZ-resistant U266 cells by immunoblotting and Annexin-V/propidium iodide staining. Immunoblotting was performed to investigate the molecular mechanism of apoptosis. Combination treatment triggered synergistic cleavage of caspase-3 and poly(ADP ribose) polymerase (PARP) in both BTZ-sensitive and BTZ-resistant U266 cells (Figure 4A). In addition, combination treatment markedly increased the cleavage of caspase-8 and caspase-9. Combination treatment markedly downregulated B-cell lymphoma-extra large protein (Bcl-xL), Bcl-2, and X-linked inhibitor of apoptosis (XIAP) antiapoptotic proteins without affecting Bax and Bak proapoptotic proteins in both BTZ-sensitive and BTZ-resistant U266 cells (Appendix A). To further confirm the apoptosis induction, we tested the Annexin-V propidium iodide staining. The population of Annexin-V-positive cells after treatment with A452 or BTZ was 18.2% and 25.8% in BTZ-sensitive and 25.2% and 15.4% in BTZ-resistant U266 cells, respectively, which increased to 54.7% in BTZ-sensitive cells and 35.0% in BTZ-resistant U266 cells, respectively, after combination treatment (Figure 4B,C and Appendix A). Furthermore, more robust results were observed following combination treatment of A452 and CFZ in both BTZ-sensitive and BTZ-resistant U266 cells (Figure 5 and Appendix A). Overall, these results indicate that A452 and BTZ or CFZ triggered apoptosis in BTZ-resistant MM cells by activating caspases and downregulating antiapoptotic factors.

### 2.5. A452 in Combination with BTZ or CFZ Synergistically Inactivates the Cell Survival Signaling in BTZ-Sensitive and BTZ-Resistant U266 Cells

HDAC6-selective inhibitors show cytotoxicity and cause apoptosis in both BTZ-sensitive and BTZ-resistant U266 cells. From these results, we hypothesized that A452 reduces BTZ resistance. Thus, we investigated the effect of combination treatment with A452 and PI on cell survival signaling in both BTZ-sensitive and BTZ-resistant U266 cells. Regardless of the resistance of BTZ, the phosphorylated levels of ERK, AKT, and p65 were decreased by A452 treatment in both BTZ-sensitive and BTZ-resistant U266 cells (Figure 6A and Figure 7A). Sensitivity to A452 in BTZ-resistant U266 cells was slightly lower than in BTZ-sensitive U266 cells. We also observed the changes in cell survival signaling by combined treatment of A452 and BTZ or CFZ. While the single treatment of BTZ did not affect the phosphorylated ERK, AKT, and p65, combined treatment of BTZ and A452 synergistically reduced the phosphorylated ERK, AKT, and p65 in BTZ-resistant U266 cells without changing total ERK, AKT, and p65 levels (Figure 6A and Figure 7A). Similar to BTZ and A452, CFZ and A452 synergistically reduced the phosphorylated ERK without altering total ERK levels in BTZ-resistant U266 cells (Figure 6B). Interestingly, CFZ treatment with A452 synergistically reduced phosphorylated AKT and phosphorylated p65 and their total forms in BTZ-sensitive U266 cells and BTZ-resistant U266 cells (Figure 6B and Figure 7B). The synergistic effect of A452 with CFZ was more potent than that of A452 with BTZ in BTZ-resistant U266 cells. Taken together, these results suggest that the HDAC6-selective inhibitor restores the sensitivity to BTZ by inactivating cell survival signaling in BTZ-resistant U266 cells.

### 2.6. A452 Enhances MM Sensitivity to BTZ or CFZ in BTZ-Resistant U266 Cells

Upregulation of the catalytically active immunoproteasome subunits LMP7 and LMP2 confers PI resistance by increasing proteasome capacity [21]. A recent study has demonstrated that LMP2 and LMP7 expression are related to STAT3 activity in BTZ-resistant MM cells [21], and NF-κB directly regulates the transcription of LMP2 [43]. We, therefore, examined whether the combination of BTZ or CFZ with A452 can overcome acquired BTZ resistance. The phosphorylated levels of STAT3 treated with A452 were more decreased in BTZ-sensitive U266 cells compared with BTZ-resistant U266 cells (Figure 7A). The BTZ-resistant U266 cells showed less sensitivity to the STAT3 and NF-κB inhibition of BTZ or CFZ compared with BTZ-sensitive U266 cells. A452 treatment with BTZ synergistically reduced phosphorylated STAT3 and phosphorylated p65 without changing the total forms of STAT3 and p65 in both BTZ-sensitive and BTZ-resistant U266 cells (Figure 7A). Inactive STAT3 and NF-κB, in turn, downregulated LMP2 and LMP7 in both BTZ-sensitive and BTZ-resistant U266 cells. Interestingly, A452 treatment with CFZ synergistically reduced both phosphorylated STAT3 and p65 and total STAT3 and p65, resulting in synergistically reduced expressions of LMP2 and LMP7 in both BTZ-sensitive and BTZ-resistant U266 cells (Figure 7B). Thus, these results indicate that A452 in combination with BTZ or CFZ could overcome BTZ-induced resistance in MM.

## 3. Discussion

Acquired anticancer drug resistance is a major problem to effective therapy for patients with MM [44]. Although anticancer drugs, such as BTZ, lenalidomide, and Dex, have exhibited clinical success, some MM patients fail to respond to these drugs, and their outcome is still poor due to primary refractories and acquisition of resistance [45]. Consequently, BTZ resistance causes many changes in the molecular character in cancer cells. In this study, we confirmed that several cell survival signaling pathways, including NF-κB, MAPK, and STAT3, were highly increased in BTZ-resistant MM cells. We showed that A452, an HDAC6-selective inhibitor, rescues the phosphorylated levels of NF-kB, ERK, and STAT3 upregulated by BTZ resistance and induces cytotoxicity in BTZ-resistant MM cells. Moreover, A452, combined with BTZ or CFZ, was shown to synergistically decrease pERK, pp65, and pSTAT3 levels and increase sensitivity to BTZ- and CFZ-induced apoptosis (Figure 8). Recent studies reported that single treatment of ACY-1215 or combination treatment of ACY-1215 and BTZ or CFZ markedly induce cell death in MM cells [27,29,31,46]. However, it is incompletely understood whether combination treatment of ACY-1215 and PI would lead to cell death in BTZ-resistant MM cells. In this study, we showed that HDAC6 inhibition remains effective in BTZ-resistant MM cells, which has not yet been explored. In addition, we demonstrated that A452, as well as ACY-1215, dose-dependently decreased the cell viability and cell growth rate of both BTZ-sensitive and BTZ-resistant MM cells. Additionally, combination treatment with A452 and BTZ or CFZ synergistically decreased cell viability and consequently enhanced apoptosis in both BTZ-sensitive and BTZ-resistant MM cells. Altogether, our findings suggest that A452 may be beneficial in the treatment of the PI-resistant refractory and relapsed MM patients.

Previous studies revealed that MAPK and AKT signaling correlates with MM cell survival [47]. AKT activates I-κB kinase and p38 MAPK, stimulating NF-κB transactivation. Activated AKT is known to synergize with co-activator CBP in the activation of the p65 transactivation domain. Phosphorylation of p65 at Ser 536 by IKK and acetylation of p65 at Lys 310 by CBP enhance its transcriptional activity [48]. A recent study showed that the inhibition of HDAC6 accumulated Lys 163 and Lys 377 on the kinase domain of AKT, consequently decreasing AKT kinase activity [49]. Consistent with previous findings, combination treatment with A452 and BTZ or CFZ synergistically inactivated AKT in both BTZ-sensitive and BTZ-resistant MM cells. Our results showed that the HDAC6-selective inhibitor inactivates the NF-κB pathway by modulating AKT in both BTZ-sensitive and BTZ-resistant MM cells.

LMP7 and LMP2 subunits of immunoproteasome directly bind to and inhibit BTZ. Immunoproteasome subunits can participate in the NF-κB pathway via the degradation of I-κB. Furthermore, the expression levels of proteasome subunits negatively correlate with sensitivity to PI [21,50]. As overexpressed immunoproteasome subunits can cause BTZ resistance [51], decreasing proteasome subunits is an important key to overcoming BTZ resistance. Previous studies focused on the relationship between BTZ resistance and LMP7 (overexpression and point mutation). Here, we found that LMP2 was overexpressed in BTZ-resistant cells. Then, we demonstrated that the combined treatment of A452 and BTZ or CFZ synergistically reduced the protein levels of LMP2 and, to a lesser extent, LMP7 in BTZ-resistant MM cells. The decreased level of LMP2 and LMP7 could lead to the accumulation of inactivated NF-κB in the cytoplasm by preventing the degradation of I-κB [10,11]. Overall, the combination of CFZ with A452 shows more potent synergistic effects in decreasing LMP2 and LMP7 than that of BTZ with A452 in BTZ-resistant MM cells. Although BTZ-resistant cells are also partly cross-resistant to CFZ, PI resistance is not universal across the different PI classes, and PIs possess drug-specific features [52,53]. We, therefore, suggest that LMP2 and LMP7 play an important role in overcoming BTZ resistance.

Next, our study explains that STAT3 has a key role in the reduction of LMP2 and LMP7 through inhibition of HDAC6. A recent study demonstrated that LMP2 and LMP7 expression levels were related to STAT3 activity in BTZ-resistant MM cells [21]. Another study has shown that NF-κB directly regulates the transcription of LMP2 by binding on the NF-κB element (GGGACTTTCC) [43]. HDAC6 also modulates activation of STAT3 [38,54] and NF-κB [38]. Our results exhibit that inactivation of STAT3 and NF-κB by HDAC6 inhibition reduces LMP2 and LMP7 in BTZ-resistant MM cells. Furthermore, these effects are synergistically enhanced when combined with A452 and BTZ or CFZ. Thus, our findings indicate that the combination of HDAC6 inhibitor and PI can overcome BTZ resistance via regulating STAT3 and NF-κB signaling pathways.

## 4. Materials and Methods

### 4.1. Reagents

ACY-1215 (ricolinostat), Bortezomib, and Carfilzomib were purchased from Selleck Chemicals (Houston, TX, USA). Powders were solubilized in DMSO (Sigma Chemical, St. Louis, MO, USA). Antibodies against AKT (sc-8312), p-AKT (sc-7985-R), Bak (sc-832), ERK (sc-03-G), HDAC6 (sc-11420), LMP2 (sc-514345), p65 (sc-8008), STAT3 (sc-482), and α-tubulin (sc-32293) were purchased from Santa Cruz Biotechnology (Santa Cruz, CA, USA). Antibodies against GAPDH (AP0066) and p-p65 (BS4138) were from Bioworld Technology (Bloomington, MN, USA). Antibody against LMP7 (A305-229A) was from Bethyl Laboratories (Montgomery, TX, USA). Antibodies against caspase-8 (#551244), caspase-9 (#551246), PARP (551024), and XIAP (610716) were from BD Biosciences (San Jose, CA, USA). Antibodies against Bax (#2772), Bcl-2 (#7382), Bcl-xL (#2762), caspase-3 (#9662), pERK (#2220), and pSTAT3 (49,138) were from Cell Signaling Technology (Danvers, MA, USA). Antibody against acetyl α-tubulin (T6793) was from Sigma-Aldrich (St. Louis, MO, USA).

### 4.2. MM Cell Lines and Culture

The BTZ-sensitive U266 and BTZ-resistant U266/VelR human MM cell lines were kindly provided by Dr. Sung-Soo Yoon (Seoul National University College of Medicine, Seoul, Korea; [34]). Cells were cultured in RPMI 1640 medium (HyClone; GE Healthcare, Logan, UT, USA) containing 10% fetal bovine serum (HyClone), 100 units/mL penicillin, and 100 µg/mL streptomycin (Gibco; Thermo Fisher Scientific, Inc., Waltham, MA, USA) in a humidified atmosphere of 5% CO_2_ and 95% air at 37 °C. BTZ-resistant U266/VelR cells were maintained in RPMI 1640 medium, as described above, containing 2 nM BTZ.

### 4.3. MM Cell Lines and Culture

To strengthen BTZ resistance, BTZ-resistant U266/VelR MM cells provided by Dr. Sung-Soo Yoon were reselected using soft agar assay. Soft agar agarose was autoclaved in distilled water before use. In total, 5 × 10^5^ BTZ-sensitive U266 cells in a 100-mm plate were spun down and resuspended in cooled 0.3% agar in RPMI 1640 medium containing 10% FBS with 10 nM BTZ. The cells were seeded onto a solidified base layer, including 1% agar in the culture media. The plates were kept at room temperature for 30 min before being placed into a CO_2_ incubator. The agar plates were incubated for 3–4 weeks until colonies were visible on the surface. Individual colonies were picked from soft agar, and pooled colonies were grown in the culture media with 2 nM BTZ [34]. We confirmed BTZ resistance in U266/VelR MM cells relative to U266 MM cells by the cell viability assay and Western blot.

### 4.4. Cell Growth and Viability Assay

Cell growth and viability were assessed by measuring a water-soluble tetrazolium salt, 2-(2-methoxy-4-nitrophenyl)-3-(4-nitrophenyl)-5-(2,4-disulfophenyl)-2H tetrazolium, monosodium salt (WST-8) (Cell Counting Kit (CCK)-8 kit, Dojindo Molecular Technologies, Inc., Kumamoto, Japan) dye absorbance. Cells were pulsed with 20 μL of WST-8 to each well for the last 3 h of 72 h, and absorbance was measured at 450 nm using a multimode microplate reader (Tecan, Männedorf, Switzerland). Results are presented as the percent absorbance relative to control cultures and were generated from three independent experiments performed in triplicate.

### 4.5. Growth Inhibitory and Viability Inhibitory Assays

Drug concentrations that inhibited 50% of cell growth (GI_50_) and 50% of cell viability (IC_50_) were determined using CCK-8 assay as described elsewhere. All cell lines were treated for 72 h on day two unless otherwise stated. GI_50_ and IC_50_ were determined using Prism Version 6.0 software (GraphPad Software, Inc., La Jolla, CA, USA).

### 4.6. Drug Combination Analysis

Synergism between PI and HDAC6 inhibitor was evaluated using the Chou–Talalay method [39]. Fraction affected (Fa) versus the combination index (CI) plot was drawn using CalcuSyn (Biofosft). The drug combination was considered synergistic when CI was less than 1.

### 4.7. Apoptosis Assay

Apoptosis was assessed using Annexin V/propidium iodide double staining according to the manufacturer (BD Biosciences, Franklin Lakes, NJ, USA). After treatments, cells were trypsinized and stained with 0.5 mg/mL Annexin V in binding buffer (10 mM HEPES free acid, 0.14 M NaCl, and 2.5 mM CaCl_2_) for 30 min. Afterward, propidium iodide (5 mg/mL final concentration) was added and incubated for another 15 min. Cells were then analyzed using a flow cytometer and BD FACSDiva software version 7 (both BD Biosciences).

### 4.8. Western Blot Analysis

Cells grown and treated as indicated were collected, lysed, and separated by sodium dodecyl sulfate-polyacrylamide gel electrophoresis (SDS-PAGE); Western blotting was performed as previously described [55]. The blots were semi-quantified using FusionCapt software version 16.08a (Viber Lourmat Sté, Collégien, France). The protein expression levels were semi-quantified relative to GADPH or α-tubulin, and the levels in the 0.1% DMSO-treated groups were set at 1. GADPH and α-tubulin were used as loading controls.

### 4.9. Statistical Analysis

All results are expressed as means ± standard deviation (SD) of three independent experiments. For the cell growth and viability test of single agents, statistical significance was determined by unpaired two-tailed Student’s *t*-test. Statistical analysis for the other data was performed by GraphPad Prism software 7.0 (Graphpad Software, San Diego, CA, USA). One-way or two-way ANOVA followed by post hoc analysis with Bonferroni’s multiple comparison test was used to evaluate statistical significance. *p* < 0.05 was considered statistically significant for the data.

## 5. Conclusions

In conclusion, our results demonstrate that PI with HDAC6-selective inhibitor induces synergistic cytotoxicity in both BTZ-sensitive and BTZ-resistant MM, associated with the inactivation of MAPK, ATK, NF-κB, and STAT3 signaling pathways. Additionally, the combination of A452 and PI synergistically reduces LMP2 and LMP7 via inhibition of the STAT3 and NF-κB in both BTZ-sensitive and BTZ-resistant MM cells. Taken together, we suggest that the HDAC6-selective inhibitor overcomes the BTZ resistance via inhibition of several different types of cell survival signaling and the STAT3–NF-κB–LMP2 pathway. Thus, our results suggest that the HDAC6-selective inhibitor may provide beneficial therapeutic opportunities for MM patients with resistance to BTZ and other PIs.

## Figures and Tables

**Figure 1 ijms-22-01341-f001:**
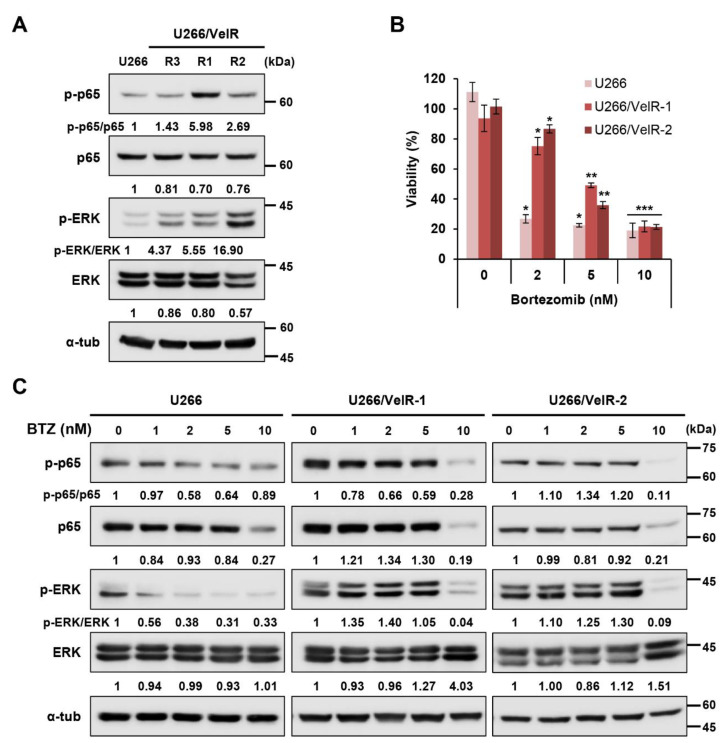
Establishment of bortezomib (BTZ)-resistant U266/VelR MM cells. (**A**) Immunoblotting analysis of both BTZ-sensitive U266 and BTZ-resistant U266 (U266/VelR-1, U266/VelR-2, and U266/VelR-3) MM cells. (**B**) Both BTZ-sensitive and BTZ-resistant U266 cells were treated with indicated concentrations of BTZ (2, 5, 10 nM) for 72 h, and cell counting kit-8 (CCK-8) assays were performed to analyze cell viability. Data represent mean ± SD (*n* = 3). Student’s *t*-test, * *p* < 0.05, ** *p* < 0.01, and *** *p* < 0.001 vs. the DMSO control. (**C**) Immunoblotting analysis of both BTZ-sensitive U266 and BTZ-resistant U266 cells treated with 0.01% DMSO or indicated concentrations of BTZ (1, 2, 5, and 10 nM) for 24 h. Relative protein expression levels were semi-quantified by densitometric analysis of the blots. α-tubulin was used as an equal loading control. The abundance of the indicated proteins was semi-quantified relative to α-tub, and control levels were set at 1.

**Figure 2 ijms-22-01341-f002:**
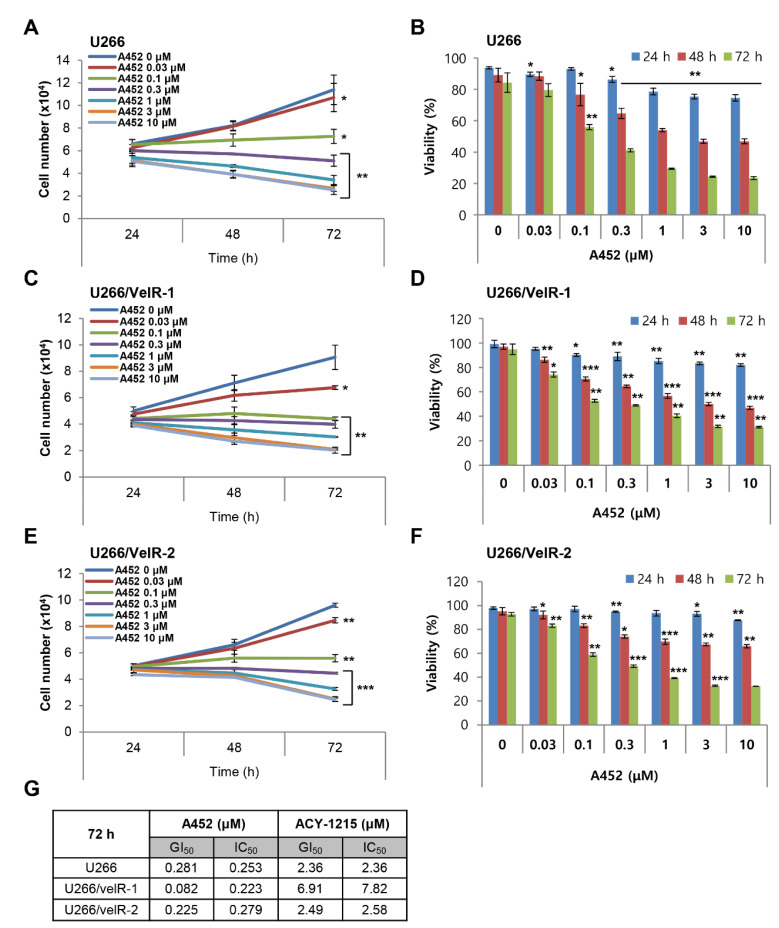
Histone deacetylase 6 (HDAC6)-selective inhibitors suppress cell growth and viability in BTZ-sensitive and BTZ-resistant U266 MM cells. (**A**,**B**) BTZ-sensitive U266, and (**C**–**F**) BTZ-resistant U266 (U266/VelR-1 and U266/VelR-2) MM cells were treated with 0.1% DMSO (control) or indicated concentrations of A452 for 72 h, and CCK-8 assays were performed to analyze cell growth and viability. Cell counts were estimated indirectly from a standard curve generated using solutions of known cell counts. Absorbance was normalized to that of the negative control at each time interval. Data present as mean ± SD (*n* = 3). * *p* < 0.05, ** *p* < 0.01, and *** *p* < 0.001 vs. the DMSO control, Student’s *t*-test. (**G**) The IC_50_ and GI_50_ values for A452 and ACY-1215 in BTZ-sensitive and BTZ-resistant U266 cells using GraphPad prism.

**Figure 3 ijms-22-01341-f003:**
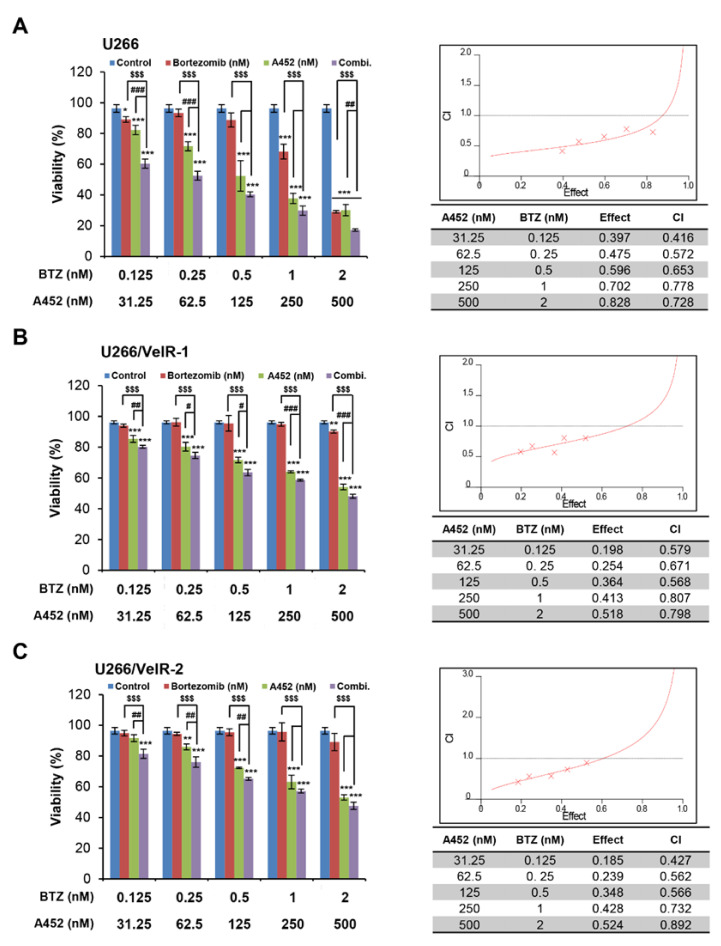
Cotreatment with A452 and BTZ triggers synergistic cytotoxicity in both BTZ-sensitive and BTZ-resistant U266 MM cells. (**A**) BTZ-sensitive U266 and (**B**,**C**) BTZ-resistant U266 (U266/VelR-1 and U266/VelR-2) MM cells were treated with 0.1% DMSO (control), A452, and BTZ or in combination with these compounds as indicated for 72 h. Combination treatments were performed in U266 (**A**) and BTZ-resistant U266 (**B**,**C**) cells maintaining a constant ratio between the dose of the A452 and BTZ, and cell viability were assessed at 72 h by CCK-8 assay. The combination index (CI) value and the relative fraction affected (FA) were determined at each dose combination (actual), and a simulation was run to estimate the CI value and confidence interval (—) across the entire FA range (simulation). CI < 1, CI = 1, and CI > 1 indicate synergistic, additive, and antagonistic effects, respectively. CI was calculated by the CalcuSyn software program. Data present as mean ± SD (*n* = 3). * *p* < 0.05, ** *p* < 0.01, and *** *p* < 0.001 vs. DMSO control, ^$$$^
*p* < 0.001 vs. BTZ-treated group, ^#^
*p* < 0.05, ^##^
*p* < 0.01, and ^###^
*p* < 0.001 vs. A452-treated group; two-way analysis of variance (ANOVA) test.

**Figure 4 ijms-22-01341-f004:**
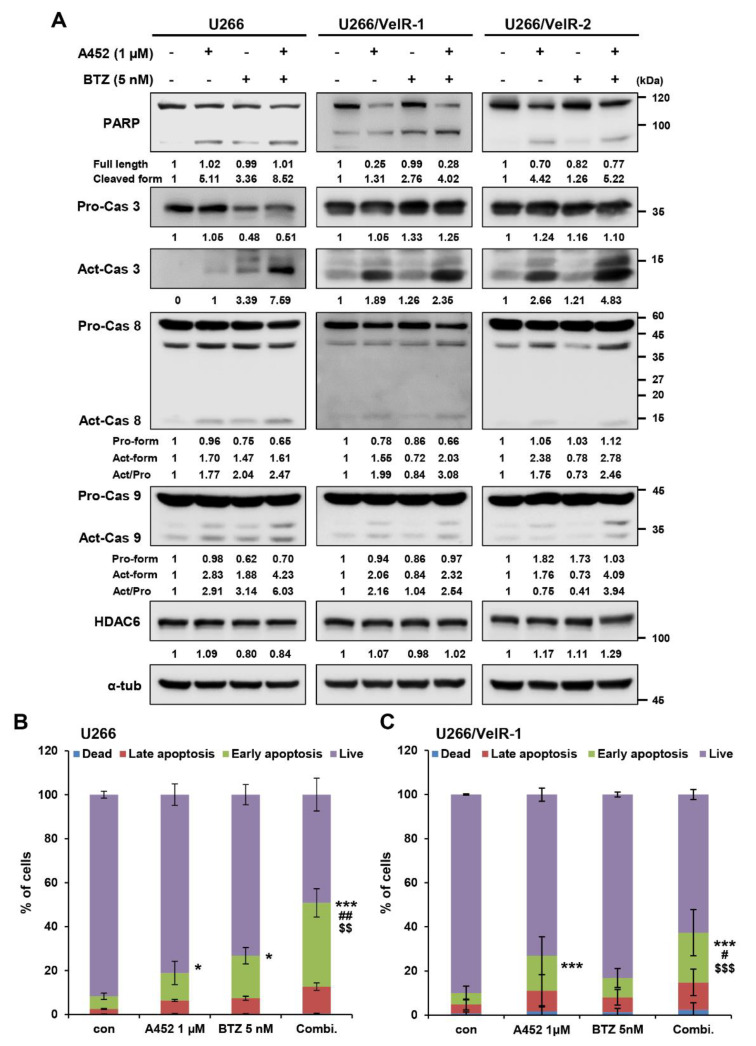
Cotreatment with BTZ and A452 leads to synergistic apoptosis induction in both BTZ-sensitive and BTZ-resistant U266 MM cells. (**A**) BTZ-sensitive U266 and BTZ-resistant U266 (U266/VelR-1 and U266/VelR-2) MM cells were treated with 0.1% DMSO (control), A452 (1 μM), and BTZ (5 nM) or in combination with these compounds as indicated for 24 h. Apoptotic markers were identified by immunoblotting using whole-cell lysates. α-tubulin was used as an equal loading control. The abundance of the indicated proteins was semi-quantified relative to α-tub; control levels were set at 1. (**B**) BTZ-sensitive U266 and (**C**) BTZ-resistant U266/VelR-1 cells were treated with 0.1% DMSO (control), A452 (1 μM), and BTZ (5 nM) or in combination with these compounds as indicated for 36 h. Cell death was assessed by flow cytometry and Annexin V/PI staining. Data present as mean ± SD (*n* = 3). * *p* < 0.05 and *** *p* < 0.001 vs. DMSO control, ^$$^
*p* < 0.01 and ^$$$^
*p* < 0.001 vs. BTZ-treated group, ^#^
*p* < 0.05 and ^##^
*p* < 0.01 vs. A452-treated group; two-way analysis of variance (ANOVA) test.

**Figure 5 ijms-22-01341-f005:**
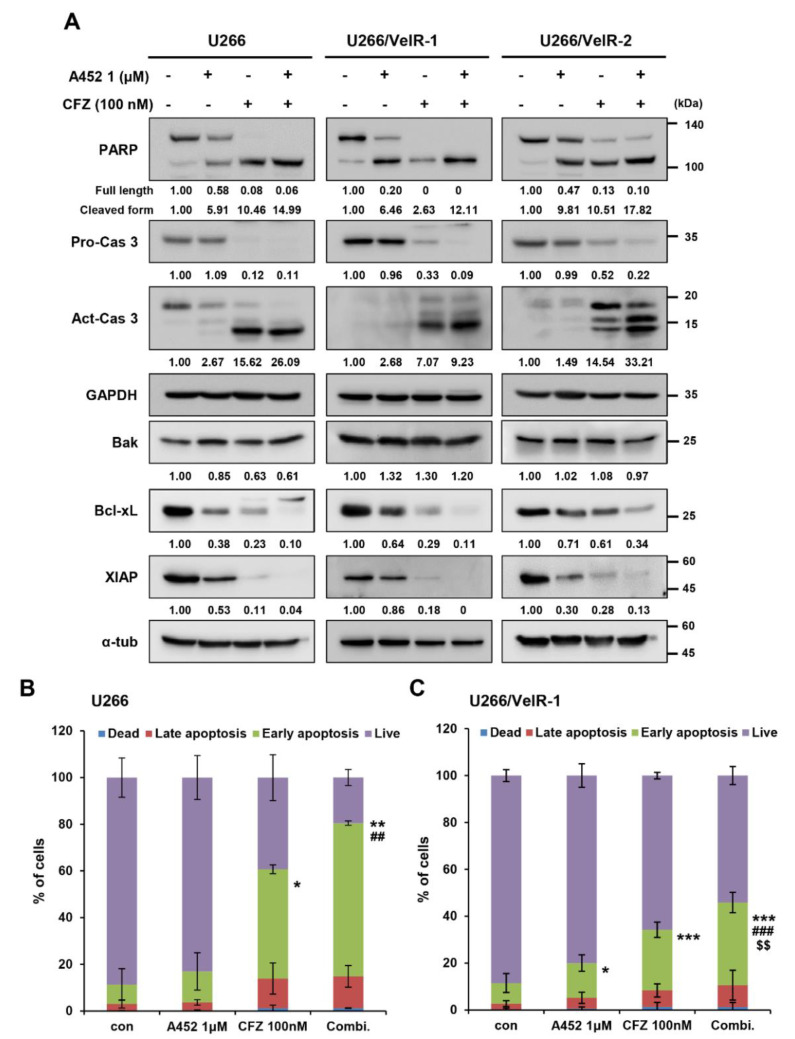
Cotreatment with CFZ and A452 leads to synergistic apoptosis induction in both BTZ-sensitive and BTZ-resistant U266 MM cells. (**A**) BTZ-sensitive U266 and BTZ-resistant U266 (U266/VelR-1 and U266/VelR-2) MM cells were treated with 0.1% DMSO (control), A452 (1 μM), and carfilzomib (CFZ) (100 nM) or in combination with these compounds as indicated for 24 h. Apoptotic markers were identified by immunoblotting using whole-cell lysates. Glyceraldehyde 3-phosphate dehydrogenase (GADPH) and α-tubulin were used as equal loading controls. The abundance of the indicated proteins was semi-quantified relative to GAPDH or α-tub; control levels were set at 1. (**B**) BTZ-sensitive U266 and (**C**) BTZ-resistant U266/VelR-1 MM cells were treated with 0.1% DMSO (control), A452 (1 μM), and CFZ (100 nM) or in combination with these compounds as indicated for 36 h. Cell death was assessed by flow cytometry and Annexin V/PI staining. Data present as mean ± SD (*n* = 3). * *p* < 0.05, ** *p* < 0.01, and *** *p* < 0.001 vs. DMSO control; ^##^
*p* < 0.01 and ^###^
*p* < 0.001 vs. A452-treated group; ^$$^
*p* < 0.01 vs. CFZ-treated group, two-way analysis of variance (ANOVA) test.

**Figure 6 ijms-22-01341-f006:**
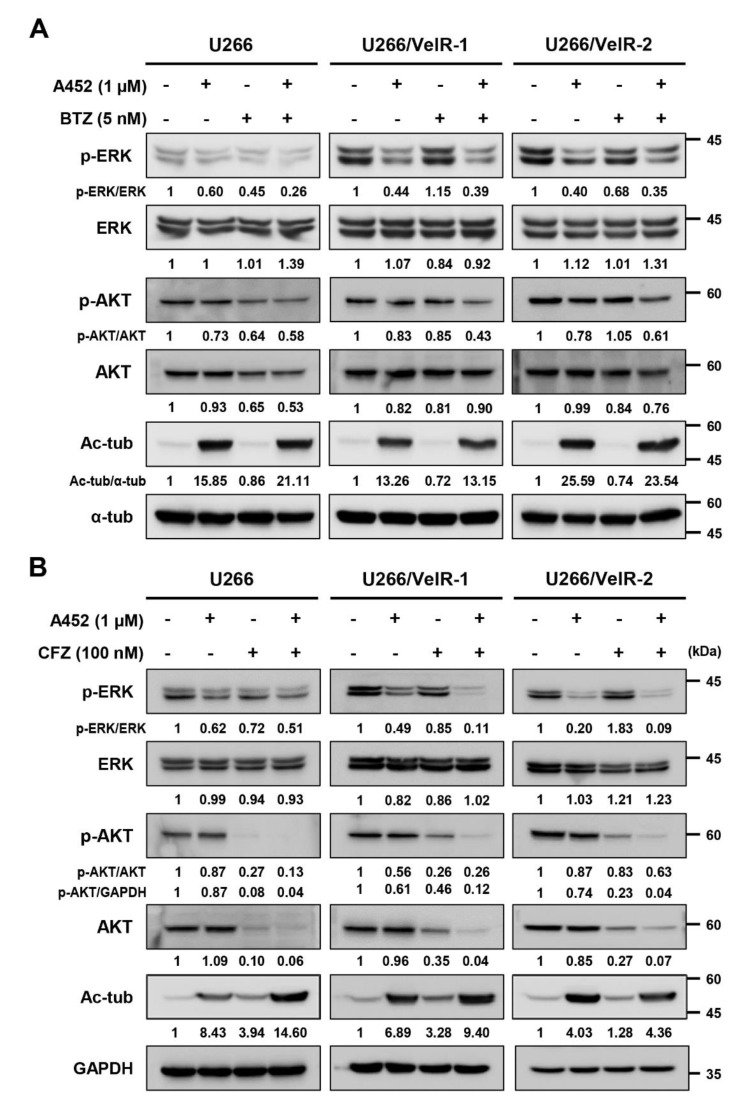
Cotreatment of A452 and PI synergistically inhibits activation of both mitogen-activated protein kinase (MAPK) and protein kinase B (PKB, known as AKT) pathways in BTZ-sensitive and BTZ-resistant U266 cells. (**A**,**B**) Both BTZ-sensitive U266 and BTZ-resistant U266 (U266/VelR-1 and U266/VelR-2) MM cells were treated with 0.1% DMSO (control), A452, BTZ (**A**), and CFZ (**B**) or in combination with A452 and PI as indicated for 24 h. Whole-cell lysates were subjected to immunoblotting with indicated antibodies. α-tubulin and GAPDH were used as equal loading controls. The abundance of the indicated proteins was semi-quantified relative to α-tub or GAPDH; control levels were set at 1.

**Figure 7 ijms-22-01341-f007:**
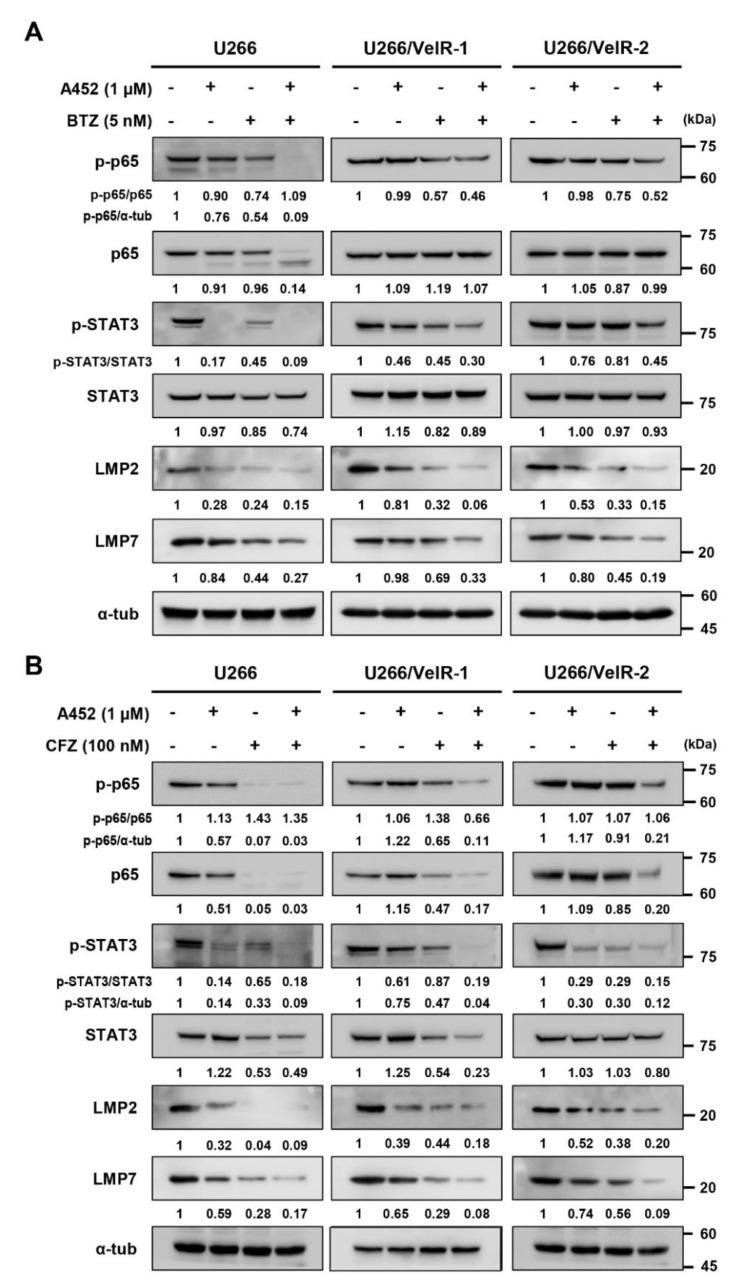
Cotreatment of A452 and PI synergistically decreases the level of low-molecular-mass polypeptide 2 (LMP2) and LMP7 via modulating signal transducer and activator of transcription 3 (STAT3) and nuclear factor kappa B (NF-κB) pathways in both BTZ-sensitive and BTZ-resistant U266 MM cells. (**A**,**B**) Both BTZ-sensitive U266 and BTZ-resistant U266 (U266/VelR-1 and U266/VelR-2) MM cells were treated with 0.1% DMSO (control), A452, BTZ (**A**), and CFZ (**B**) or in combination with A452 and PI as indicated for 24 h. Whole-cell lysates were subjected to immunoblotting with indicated antibodies. α-Tubulin was used as an equal loading control. The abundance of the indicated proteins was semi-quantified relative to α-tub; control levels were set at 1.

**Figure 8 ijms-22-01341-f008:**
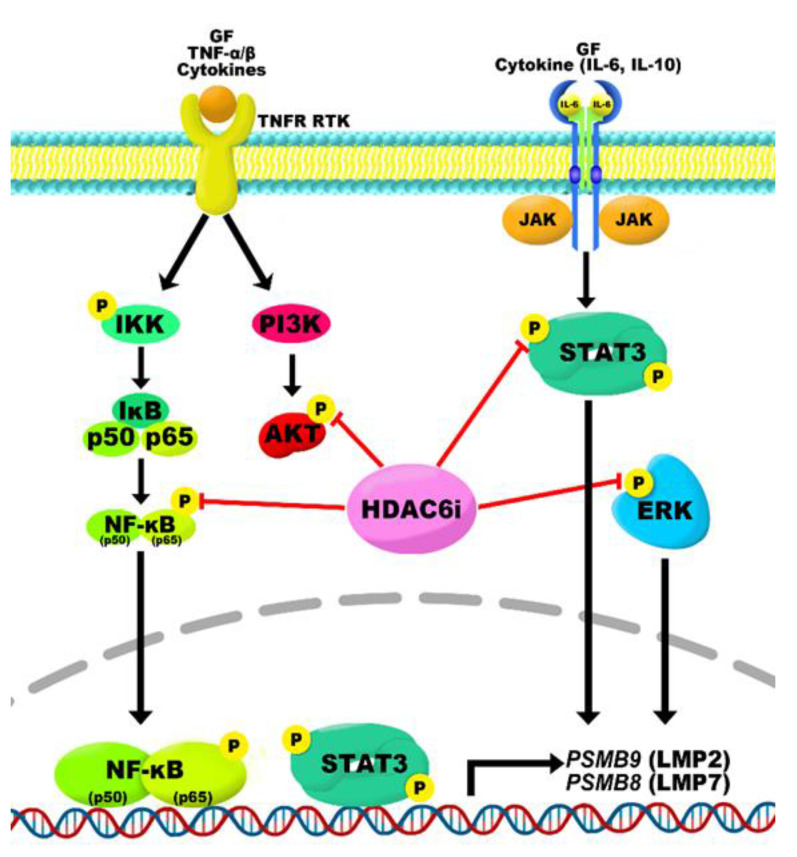
Working model for HDAC6 inhibitor and PI for overcoming BTZ resistance of MM. HDAC6 inhibition by A452 synergize with BTZ or CFZ to inhibit the activation of NF-κB and STAT3, resulting in decreased expressions of LMP2 and LMP7. Additionally, combination treatment of A452 and BTZ or CFZ synergistically inactivates the AKT and ERK MAPK pathways. Combining A452 with BTZ or CFZ leads to synergistic cancer cell growth inhibition and viability decreases. This combination induces apoptosis and enhances PI sensitivity in the BTZ-resistant MM cells.

## Data Availability

Data is contained within the article or Appendix A.

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
