# Peer review of "HDAC6-Selective Inhibitor Overcomes Bortezomib Resistance in Multiple Myeloma"

_ijms, 2021, doi:10.3390/ijms22031341_

Round 1

Reviewer 1 Report

In this work, authors investigate the role of A452 HDAC6 inhibitor in overcoming bortezomib resistance in multiple myeloma. BTZ resistant markers expression is evaluated upon treatment with A452 alone or in synergy with BTZ or CFZ in BTZ-resistant MM cells. Moreover, cell viability and apoptosis induction are assessed. Authors conclude that synergistic effect of A452 with CFZ is more potent than that of A452 with BTZ in BTZ resistant MM cells.

Although the focus is clearly of interest for the scientific community and for its potential therapeutic application, some points need to be addressed.

Following, corrections, together with questions that would need to be answered, clarified and explained to make the results understandable, are listed.

Overall, I expect that, upon major revision, the manuscript will be suitable for publication.

  • Line 44: correct with “signaling”
  • Line 67: correct with “the degradation of”
  • Line 94: correct with “with”
  • Line 106: Vel in capital letters
  • Line 107: hyper
  • Line 130: GI50 and IC50 lowercase letters
  • Line 136: IC50 lowercase letters
  • Line 176: remove “then”
  • Line 199: blood cancers: which combination? Please, explain better referring also to “hematological malignancies” reported in the following sentence
  • Line 244: correct with “cause”
  • Line 307-309: sentence is not clear
  • Line 326-327: sentence is not clear
  • Line 435: explicit with “Supplementary Figure 1”
  • Along the text: check if Student’s t test is properly written
  • Figure 1B: U266/Vel-R2 in capital letters
  • Figure 2C: ***stars are not aligned with the histograms
  • Figure 2G: remove table regarding ACY1215 data

  • There are many differences and discrepancies in the drug concentrations used for the different experiments described. It would be necessary to clearly explain and justify in the text how and why each specific one has been chosen. This is a summary of what it is reported in the different experiments.

Concentrations of drugs used:

BTZ alone: range 0-10 nM (Figure 1)

A452 alone: range 0-10 µM (Figure 2)

BTZ+A452 combined: BTZ range 0-2nM, A452 range 31-500 nM (Figure 3)

BTZ+A452: BTZ 5nM, A452 1 µM (Figure 4, 6, 7, S4)

CFZ+A452: CFZ 100nM, A452 1 µM (Figure 5, 6, 7, S5)

CFZ alone: range 0-100 nM (Figure S2)

CFZ+ A452 combined: CFZ range 1-100 nM, A452 range 31-2000 nM (Figure S3)

  • Figure 1: explicit why it has been chosen to perform experiments in U266/Vel-R1 and U266/Vel-R2 and not in U266/Vel-R3.

  • Figure 4C, Figure 5, Figure S4, Figure S5: U266/Vel-R2 results are not reported: why?
  • Figure 4 and Figure 5: Annexin/PI analyzed at 36 hours
  • Figure S4, Figure S5: Annexin/PI analyzed at 24 hours

Why this timing difference has been chosen?

  • Figure 4: explicit all ratios, for each protein analyzed.
  • U266/Vel-R1: PARP cleavage with A452 alone and in combination is very similar, how can it be justified?
  • U266/Vel-R1: Cas8 cleavage with A452 alone and in combination is very similar, how can it be justified?
  • U266/Vel-R1: Cas9 cleavage with A452 alone and in combination is very similar, how can it be justified?
  • U266: Cas3 cleavage with A452 alone and in combination is very similar, how can it be justified?
  • Explain why HDAC6 protein has been used in this experiment.

  • Figure S4:
  • U266/Vel-R1: BAX in combination decreases, how can it be justified?
  • U266/Vel-R1, U266/Vel-R2: Bcl 2 decreases with BTZ alone and in combination, how can it be justified?
  • U266/Vel-R1 and U266/Vel-R2: Bclxl decreases with A452 alone and in combination, how can it be justified?

  • Figure 5: explicit all ratios, for each protein analyzed.
  • U266/Vel-R1: PARP cleavage with CFZ alone and in combination is very similar, how can it be justified?
  • U266/Vel-R1: XIAP decreases with CFZ alone and in combination, how can it be justified?
  • Cas8, Cas9, Bax, Bcl2 western blots are missing in the figures.

  • Figure 6, explicit why Ac-Tub has been used in this experiment.
  • Line 247-249: does the sentence refer to the A452 alone treatment?
  • Figure 6A: results for AKT are not clear, please detail the quantification of total AKT as it has been done for ERK.
  • U266/VelR-2: p-ERK decreases in presence of BTZ, how can it be justified?
  • U266/VelR-1: p-AKT decreases in presence of BTZ as much as with A452 alone, how can it be justified?
  • Figure 6B: why is AKT/GAPDH quantification detailed only in this case?
  • U266: total AKT and p-AKT decrease when treated with CFZ alone, how can it be justified?

  • Figure 7A, p65 and p-p65 data are not clear; p-p65/alpha tub, why is it reported only for U266 cells? The decrease in p-p65 in presence of A452 is minimal both in U266 and U266/Vel-R cells, how can it be justified?
  • Line 267: It is stated that “p65 treated with A452 were more decreased in BTZ-sensitive U266 cells compared with BTZ-resistant”, but it does not seem so from the results.
  • Line 272: you state that “Inactive STAT3 and NF-κB, in turn, downregulated LMP2 and LMP7 in both BTZ-sensitive and BTZ-resistant U266 cells”. However, NF-kB has not been analyzed in this work.

Reviewer 2 Report

The paper is interesting and timely. Both BTZ and CFZ are very potent drugs and revealing the mechanism of resistance as well as identifying synergistic combinations is of high importance for both MM and other cancers such as lymphoma. 

the paper is detailed and clear to the reader and i have no comments or changes. 

Well done!

Author Response

We appreciate the reviewers for recognizing the potential importance of our findings. We believe that your comments will improve further our research on BTZ resistance greatly.

Reviewer 3 Report

Dr. Lee and colleagues present studies focused on combinatorial use of HDAC6 inhibitor A542 and bortezomib (BTZ) in pre-clinical model of BTZ-refractory multiple myeloma (MM). The authors find that A542 could antagonize malignant growth of BTZ-resistant MM cell line and could act synergistically with proteasome inhibitor BTZ or CFZ in this setting.

Major points:

  1. The manuscript lacks novelty. The use of HDAC6 inhibitors to treat BTZ-refractory MM, alone or in combination with BTZ, has been extensively reported by other groups (Fi H, Front Oncol, 2019; Sun X, Biosci Rep, 2019; Lee HY, J Med Chem, 2018; Hideshima T, PNAS, 2016; Santo L, Blood, 2012; Hideshima T, PNAS, 2005). HDAC6 inhibitors have also been described in combination with CFZ (Mishima Y, Br J Haematology, 2015).
  2. It is not clear if A542 could be used in the clinical setting, and in vivo studies have not been reported in this manuscript,  pointing to poor translational significance
  3. The use of only 1 MM cell line does not reflect the clonal heterogeneity featuring MM and it is not acceptable.

Round 2

Reviewer 1 Report

I would like to thank the authors for having provided extensive responses to all the questions I raised. The work has been fully improved and answers have been exhaustive.

I have only a few minor points to be completed before full acceptance of the manuscript:

  • Line 44: correct with “signaling”.
  • Line 146, line 171, 453, 457 correct with "Student’s" (apostrophe is missing).
  • Figure S4: I don’t see data modifications related to Bclxl in the new Figure.
  • Please update the "Supporting Information.pptx" file with the new western blot panels inserted in the revised manuscript.

Author Response

I would like to thank the authors for having provided extensive responses to all the questions I raised. The work has been fully improved and answers have been exhaustive.

I have only a few minor points to be completed before full acceptance of the manuscript:

  • We thank for your revision. If your comments were absent, this paper would not have been complete. Also, we are delighted to know that you are satisfied with our responses.
  • Line 44: correct with “signaling”. -> Thank you for the note. We have corrected it.
  • Line 146, line 171, 453, 457 correct with "Student’s" (apostrophe is missing). -> Thank you for your comment, we have corrected it.
  • Figure S4: I don’t see data modifications related to Bclxl in the new Figure.  -> As for figure S4, we apologize for the confusion. We did not make any corrections to Figure S4 from the last rebuttal. We did not make additional corrections because as shown in Figure 3, both U266/VelR-1 and U266/VelR-2 showed synergistic decrease in cell viability when A452 and BTZ was treated together, compared to either drug treated alone. Also, we showed through FACS analysis (Apoptosis assay) in Figure 4 that synergistic apoptosis occurs when A452 and BTZ is used in combination in U266/VelR-1. Moreover, our western blot analysis shows that other anti-apoptotic markers such as Bcl-2 and XIAP significantly decreased upon A452 and BTZ combination. As a result, like the reviewer noted, although Bcl-xL decrease less dramatically in synergy compared to single treatment of A452 and BTZ, other anti-apoptotic markers as well as other experiments indicate clear synergy between A452 and BTZ. Therefore, although the protein expression level of Bcl-xL was only slightly more decreased in combination treatment than in A452 single treatment, we concluded that the figures does not need further revision. We hope this response satisfies you.
  • Please update the "Supporting Information.pptx" file with the new western blot panels inserted in the revised manuscript. -> We thank you for your kind comment. We submit “Supporting Information.pptx” to you and the editors.

Reviewer 3 Report

Unfortunately, I am not convinced by author's reply. In my opinion the papers still lacks novelty (very far to describe the molecular mechanism of BTZ-resistance) and translational value. The molecular mechanisms are very poorly described, too.

Author Response

  • We regret that our responses failed to satisfy you. In following papers, we hope to conduct further studies based on your comments. Thank you for your consideration.